

# SMAD-LDS: enhanced secure message authentication and dissemination with lightweight digital signature in the Internet of Vehicles

Islam Z. Ahmed[1], Yasser Hifny[1] and Rowayda A. Sadek[1,2]

[1] Faculty of Computers and Artificial Intelligence, Helwan University, Cairo, Egypt
[2] Saxony Egypt University for Applied Science and Technology, Cairo, Egypt

Corresponding authors
Islam Z. Ahmed,
islam.zakaria@fci.helwan.edu.eg
Rowayda A. Sadek,
rowayda_sadek@yahoo.com

## ABSTRACT

The Internet of Vehicles (IoV) plays a crucial role in enhancing driver experience and road safety by enabling vehicles to share real-time traffic information. However, the nature of the IoV communication exposes vehicles to potential security risks. Therefore, robust authentication mechanisms are essential to prevent malicious vehicles from disseminating misleading information. Ensuring secure communication in IoV necessitates addressing critical security and privacy requirements, including entity and data authentication, privacy preservation, and confidentiality. Recent research in this area frequently encounters challenges related to high computation and communication overheads, as well as vulnerability to diverse security threats. Consequently, this article proposes secure message authentication and dissemination with lightweight digital signature (SMAD-LDS), a robust message authentication and dissemination scheme incorporating a novel lightweight digital signature technique designed to minimize these overheads. There are four main phases in the proposed SMAD-LDS scheme such as the initiation phase, registration phase, send/receive message phase, and key updating phase. For the registration phase, our results demonstrate that the computation cost in SMAD-LDS is reduced by at least 46.8% compared to other works in the literature. Moreover, the computation overhead in the send/receive messages phase in SMAD-LDS is reduced by at least 94%. Additionally, the overall cost of communications in the proposed article has improved. The cost of communications in the registration phase and send/receive message phase has improved by 28% and 70%, respectively, compared to other works. Furthermore, SMAD-LDS is resistant to known attacks such as modification attacks, impersonation attacks, eavesdropping attacks, fake roadside unit (RSU) attacks, traceability attacks, and replay attacks.

## INTRODUCTION

Nowadays, human life can't be imagined without vehicles. However, the rise in traffic accidents is a significant issue with vehicles and the transportation system. According to the World Health Organization (WHO), about 1.3 million people annually die due to road traffic crashes. Moreover, traffic crashes in most countries cost around 3% of the gross

domestic product (*World Health Organization, 2023*). There are many reasons for these accidents, such as driver mistakes, lack of awareness, insufficient traffic signs, and bad road design. These accidents highlight the urgent need to find solutions for saving human lives. The intelligent transportation system (ITS) may be the solution to these issues. The objectives of the ITS are to enhance road safety and alleviate congestion through effective traffic flow management.

VANET is a vehicular *ad hoc* network that allows wireless communication between vehicles and infrastructure along the road. VANET nodes possess specific components that facilitate the collection of road information. These components include sensors, cameras, GPS, and omnidirectional antennas (*Prasan & Murugappan, 2016*; *Siddiqui, Khaliq & Pannek, 2017*). VANETs play a crucial role in the ITS by enabling vehicles to act as mobile sensors and communication nodes, collecting, disseminating, and routing information about traffic conditions.

VANET has three main components, such as the on-board unit (OBU), roadside unit (RSU), and trusted authority (TA). OBU is a wave device mounted on each vehicle. It is responsible for exchanging information between vehicles and RSUs *via* dedicated short-range communication (DSRC) (*Azam et al., 2021*). RSUs are fixed units installed by the TA in dedicated locations along the road. All RSUs along the road can communicate with each other (*Shetty & Manjaiah, 2022*). The TA manages the entire VANET system and is responsible for the OBUs and RSUs.

Traditional VANETs primarily utilized vehicle-to-vehicle (V2V) and vehicle-to-infrastructure (V2I) communication models (*Afzal & Kumar, 2020*). However, recent advancements necessitate additional models, including vehicle-to-sensor (V2S), vehicle-to-pedestrian (V2P), and vehicle-to-network (V2N) communication (*Hakimi et al., 2021*; *Yoshizawa et al., 2023*). To accommodate these evolving needs, a new VANET architecture, the Internet of Vehicles (IoV), has been proposed. The IoV enables vehicles to communicate with devices, people, and diverse networks, thereby enhancing the security and safety of intelligent transportation systems (ITS) (*Verma & Sharma, 2021*; *Sharma & Sharma, 2020*).

There are many distinguishing characteristics of the IoV, such as high and predictable mobility of vehicles, frequently disconnected networks, geographically constrained, highly dynamic topology, large network scale, sufficient energy (no power constraints), high computational resources, real-time communication, variable density, integration with other networks, and diverse applications (*Hasrouny et al., 2017*; *Hozouri et al., 2023*).

The IoV has the potential to enable a diverse range of applications, which can be broadly categorized into two main groups: safety-related and entertainment-related (*Alalwany & Mahgoub, 2024*; *Tufail et al., 2021*; *Elsagheer Mohamed et al., 2022*). Below are some examples:

- **Collision avoidance:** This system warns the driver of an impending collision with a vehicle in front by using sensors like radar and cameras. The IoV enhances this capability by enabling real-time data sharing about the leading vehicle's braking status with following vehicles, providing earlier and more precise alerts.

- **Blind spot detection (BSD):** This system detects vehicles in the driver's blind spots. The IoV enhances this feature by enabling data exchange about vehicles in the blind spots of nearby cars, extending the detection range, and significantly improving situational awareness.

- **Road hazard warnings:** This application allows vehicles to report road dangers like potholes, accidents, or slippery conditions to a central network. This information is then relayed to other vehicles in the vicinity, enabling drivers to take necessary precautions. Real-time updates on road conditions significantly improve driving safety.

- **Driver drowsiness detection:** This application can play a crucial role in mitigating the risks of driver exhaustion by using sensors to monitor driver behavior, such as eye tracking and steering patterns. Upon detecting signs of drowsiness, these applications can issue timely alerts or recommend rest stops, thereby enhancing road safety.

- **Real-time traffic information for navigation:** IoV plays a key role in traffic management by delivering real-time traffic information to navigation systems. This allows drivers to select the most efficient routes, avoiding congestion and reducing travel time.

- **Connected car apps:** IoV enables a wide range of connected car apps that provide information about nearby points of interest, restaurants, gas stations, and parking availability. These apps enhance travel convenience.

- **Integration with smart home:** These applications can connect with smart home devices, enabling drivers to manage home appliances remotely. This could include actions such as pre-heating the house or turning on lights before reaching home.

To ensure the safe and reliable operation of these applications, a robust and comprehensive security scheme is necessary. This scheme must fulfill all necessary security requirements, such as authenticity, confidentiality, anonymity and untraceability, integrity, non-repudiation, and resistance against well-known attacks (*Yu et al., 2020*). Moreover, this scheme should operate swiftly with low communication and computation overhead due to the stringent time constraints in most IoV applications. Without a robust scheme, attackers could physically or financially harm drivers and access sensitive information.

The security of the IoV depends critically on ensuring message authentication and data integrity. Verifying the authenticity of the message originator is essential, as the trustworthiness of information depends on the sender (*Siddiqui et al., 2021*; *Hbaieb, Ayed & Chaari, 2022*). The absence of effective and robust vehicle authentication in the IoV can lead to significant and varied problems, such as:

- The safety of critical vehicle applications, including collision avoidance, lane change warnings, and pedestrian detection, is threatened when malicious vehicles disrupt the necessary V2X (Vehicle-to-Everything) communication.

- Malicious vehicles could be able to send false routing information, creating chaos and increasing the risk of accidents

- Malicious vehicles can compromise the privacy of drivers and passengers. It can access sensitive driver information, such as personal data and financial information.
- The economic impact of malicious vehicles could be substantial, including damage to vehicles and infrastructure, increased insurance costs, and the potential for loss of life.
- A malicious vehicle will be able to impersonate another legitimate vehicle or even an RSU.

Nevertheless, implementing effective message authentication in IoV faces significant challenges, including computational and communicational limitations, real-time constraints, network scalability, diverse security threats, privacy concerns, and key management complexities. To overcome these challenges, a robust and efficient authentication scheme must be developed. This article proposes a secure message authentication and dissemination scheme with a lightweight digital signature (SMAD-LDS), which is designed for low computation and communication overhead. The main contributions of the SMAD-LDS scheme are as follows:

- The proposed scheme provides a novel lightweight message signature technique to meet the strict real-time message dissemination requirements in the IoV. This innovative technique reduces computation and communication overheads for fast authentication, ensuring timely message processing and delivery.
- The proposed scheme comprehensively addresses key security objectives. It provides message authentication with privacy preservation, ensuring verifiable origin without revealing sender details. Furthermore, it guarantees message integrity and non-repudiation, preventing unauthorized modifications, and providing proof of origin. Message confidentiality is maintained, protecting content from unauthorized access. Additionally, message freshness mitigates replay attacks by verifying message recency. Finally, access control restricts dissemination to authorized entities.
- This proposed scheme provides strong protection against a variety of common threats. It effectively safeguards against message tampering, ensuring the integrity of data. Additionally, it prevents impersonation, allowing only authorized entities to participate. Confidentiality is maintained, protecting against eavesdropping. The system also counters fake RSU attacks, stopping malicious entities from impersonating an RSU. Finally, it prevents replay attacks, ensuring that reused messages do not undermine security.
- The performance analysis demonstrates that our scheme significantly enhances both computation and communication overheads compared to existing state-of-the-art schemes. Additionally, experimental results show that the proposed scheme reduces computation overhead by approximately 94% and communication overhead by around 70% relative to other schemes.

This article is organized as follows: "Related Work" reviews recent related work. "The Proposed SMAD-LDS Scheme" presents a detailed explanation of the proposed SMAD-LDS scheme. "Performance Analysis" evaluates the scheme's

performance. "Analysis of Security Requirements and Attack Countermeasures for the SMAD-LDS Scheme" examines the scheme's security, requirements, and countermeasures. Finally, "Conclusions and Future Works" concludes the article and outlines directions for future research.

## RELATED WORK

Authentication is considered the first step of security in the IoV. There are two phases of authentication: the message authentication phase and the data authentication phase. In the IoV, when a vehicle becomes a part of a network, it must authenticate itself at the TA. Consequently, the TA can differentiate between legitimate and malicious vehicles. In addition, when a message is transmitted between two nodes, this message must be authenticated to ensure its integrity (*Talpur & Gurusamy, 2021*).

Numerous schemes exist for message authentication with privacy preservation in the IoV, as depicted in Fig. 1. All researchers in the literature are focused on enhancing cryptography algorithms, verification processes, or signature processes. In this section, we briefly present a few existing message authentication schemes.

*Jahanian, Amin & Jahangir (2015)* proposed a TESLA protocol that first broadcasts the encrypted message with its message authentication code (MAC). Then, after a specified time, broadcast the encryption key to allow receivers to verify the message's authenticity. This scheme is not suitable for safety applications that have critical time constraints. TESLA++ is an enhanced version of TESLA, offering improved efficiency and security. It operates by initially broadcasting a MAC followed shortly after by the complete message and the corresponding authentication key (*Manvi & Tangade, 2017*).

*Lim & Manivannan (2016)* proposed an efficient authentication protocol. This protocol depends on the vehicle-to-infrastructure (V2I) communication model. Moreover, they used the Diffie–Hellman (DH) protocol to establish and share a symmetric key. They use both symmetric key and asymmetric key cryptography algorithms. *Liu et al. (2016)* enhanced the message dissemination technique of *Afzal & Kumar (2020)* by using an aggregate message authentication code instead of the onion signature.

*Ahmed, Mohamed & Sadek (2017)* proposed a low computation overhead protocol that uses CRT-RSA cryptography algorithm instead of using the traditional RSA. This algorithm can decrypt the message faster than the traditional RSA. Moreover, this protocol generates a shared symmetric key without using the well-known Diffie-Hellman protocol, which produces more computation overhead at each node. *Mohamed, Ahmed & Sadek (2021)* enhanced the authentication scheme of Islam (*Ahmed, Mohamed & Sadek, 2017*) by combining symmetric and asymmetric cryptography techniques efficiently.

*Jiang, Ge & Shen (2020)* proposed an anonymous authentication scheme based on a group signature (AAAS). This protocol combines the group signature and pseudonym mechanism to prevent a single authority from figuring out the real identity.

*Goudarzi et al. (2022)* proposed a secure and privacy-preserving authentication scheme. This scheme depends on a fog server to leverage the system throughput by moving part of the computational power to the edge of the network. Moreover, it utilizes the quotient filter

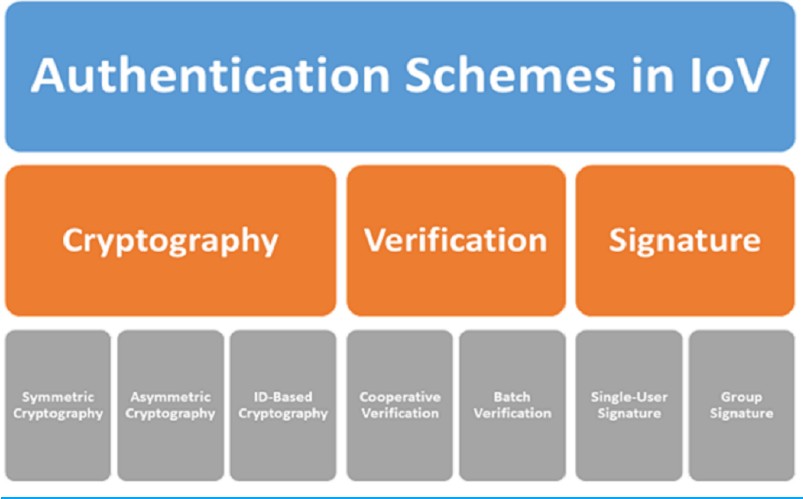

**Figure 1 Authentication schemes in the IoV.**     

(QF) to address node authentication and elliptic curve cryptography (ECC) for message authentication.

*Mistareehi & Manivannan (2022)* proposed a low-overhead message authentication and dissemination scheme. The RSU's region in this scheme is divided into sub-regions (groups). The RSU appoints group leaders who oversee each sub-region. The group leader is responsible for gathering messages from vehicles, verifying their authenticity, compiling them, and sending them to the RSU. This scheme reduces the message authentication overhead at the RSU.

*Li, Yang & Chen (2023)* proposed a privacy authentication scheme with exculpability. This model applied a double-key approach to achieving trust communication between vehicles and the RSU. Moreover, it combined the group-based methods with identity-based methods. This combination enhanced the message authentication process.

*Genc et al. (2023)* proposed a novel identity-based privacy-preservation anonymous authentication scheme (NIBPA). The author focused on the vehicle-to-vehicle communication model. Moreover, this scheme utilized pairing-free elliptic curve cryptography (ECC) to minimize the computation cost. In addition, they applied batch message verification to improve the overall performance.

*Naskar et al. (2024)* introduced a distributed authentication and revocation scheme for Vehicle-to-Infrastructure (V2I) communication. Their protocol employs a hybrid approach, combining symmetric and asymmetric cryptography. Communication occurs in discrete time sessions, or epochs, each secured by a unique epoch key. Within each epoch, authenticated message dissemination among verified vehicles is permitted. Additionally, vehicles can report misbehavior to Roadside Units (RSUs).

*Gu et al. (2022)* proposed a multi-fog-based authentication architecture designed to reduce overall delay by enabling vehicles to be verified by fog nodes. However, their protocol's centralized nature creates a communication bottleneck at the Certificate Authority (CA).

*Cui et al. (2020)* proposed a secure, privacy-preserving authentication scheme where each vehicle receives an internal pseudo-identity (IPID) from the Trusted Authority (TA), derived from its real identity. The vehicle stores this IPID, along with a self-chosen encryption key, in a tamper-proof device (TPD). Following authentication with the TA, the vehicle generates a public pseudo-identity (PPID) used for signing transmitted messages, which are then verified by the receiver. While the scheme enhances security through periodic updates to both the IPID and encryption key, it may introduce challenges related to scalability and the management of pseudo-identities in large-scale networks.

*Manivannan, Moni & Zeadally (2020)* proposed a secure privacy-preservation authentication scheme utilizing ID-based cryptography to reduce concerns with certificate management. Moreover, it employs identity-based batch signature verification for efficient RSU verification of multiple signatures. However, this scheme is susceptible to various attacks, including replay attacks.

*Chen & Chen (2021)*, proposed a certificateless aggregate signature (CLAS) scheme that simplifies the authentication process by eliminating certificate management. It offers conditional privacy, enabling vehicles to use pseudonyms for message signing to protect real identities while preserving traceability by a trusted authority. The aggregate signature mechanism further enhances verification efficiency by combining multiple signatures into a single compact signature. However, unlinkability is not a feature of this scheme.

*Badshah et al. (2022)* present an anonymous authenticated key exchange scheme for blockchain-enabled Internet of Vehicles in smart transportation (AAKE-BIVT). It utilizes lightweight encryption and physical unclonable function (PUF) to reduce computational overheads. In this scheme, vehicles are organized into clusters, and each cluster has a cluster head that communicates directly with the RSU. Moreover, it utilizes a blockchain for decentralized trust management. These techniques suffer from high computational overhead and are unsuitable for real-time applications.

To conclude, an efficient message authentication scheme should satisfy the privacy preservation condition. Moreover, it should incur low computation and communication overhead. In addition, it must resist different types of known and unknown attacks. Accordingly, this article presents an enhanced secure message authentication and dissemination scheme with a novel lightweight digital signature (SMAD-LDS).

## THE PROPOSED SMAD-LDS SCHEME

This section provides a detailed description of the proposed lightweight message authentication and dissemination (SMAD-LDS) scheme.

The scheme is designed to minimize both the computational burden and communication costs while providing robust security. The SMAD-LDS employs a regional architecture wherein the entire area is divided into distinct regions (groups), each under the control of an RSU. The Vehicle-to-Infrastructure (V2I) is the main communication model. Therefore, all communications must be directed through the RSU. Consequently, the RSU is responsible for verifying the authenticity and integrity of each message before broadcasting it to the group. This communication model is more effective regarding the

cost of communication since the RSU verifies the message originator's authenticity only once.

### The objectives of the SMAD-LDS scheme

Existing message dissemination schemes demonstrate excessive computation and communication overheads. Consequently, this article introduces the SMAD-LDS scheme, an enhanced lightweight secure message dissemination scheme designed to mitigate this shortcoming, with the following objectives:

- **Entity and data authenticity**. It is important to verify the authentication of all entities before sending and receiving messages. Moreover, it is important to prove the authenticity of the received data (messages).
- **Privacy preservation**. In the IoV, many attacks aim to track vehicles to steal private information. This information is such real identity and other sensitive personal data. Thus, it is crucial to hide the real identity of the vehicles to protect and preserve their privacy.
- **Message confidentiality**. It is important to protect and encrypt the content of the messages. This process ensures that unauthorized vehicles cannot inspect the transmitted messages' content.
- **Resistance to security attacks.** The proposed scheme should encounter all well-known attacks such Sybil attack, Masquerading attack, Eavesdropping attack, Replay attack, ID disclosure attack, Man in the middle attack, and Repudiation attack.
- **Low computation and communication overhead.** It is essential for an efficient scheme to authenticate and disseminate messages swiftly with less computation and communication overhead.

### The proposed assumptions

Many assumptions have been made to present and test the SMAD-LDS scheme. These assumptions are as follows: The proposed scheme assumes a trusted TA and RSUs, invulnerable to attacks, with secure wired or wireless communication between them. Also, vehicle and RSU security parameters are periodically updated. In addition, every vehicle is preloaded with the location, IDs, and public keys of all RSUs. Moreover, the operational whole area is divided into adjacent groups, each managed by an RSU that controls all intra-group communication, primarily V2I. Furthermore, vehicles entering an RSU's region can directly receive its messages. In addition, there is a secure routing algorithm that facilitates message forwarding from vehicles to the RSU. In addition, there is a robust malicious vehicle detection and certificate revocation mechanism.

### The SMAD-LDS scheme

The SMAD-LDS scheme has four main phases: the Initiation phase, the Group Registration Phase, the Send/Receive a Group Message, and the Leave and Join another Group Phase.

Table 1 lists all notations used in this article.

| Table 1 Notations and abbreviations. | |
|---|---|
| **Notation** | **Description** |
| $R$ | An RSU |
| $V$ | A Vehicle |
| $TS$ | A Timestamp |
| $X_{pos}, Y_{pos}$ | The Location coordinates $(X, Y)$ |
| $M$ | A Message |
| $Cert_V$ | The Vehicle's Certificate |
| $ID_v$ | The Vehicle's real identity |
| $PID_v$ | The Vehicle's pseudo-identity |
| $H()$ | Hash Function |
| $E()$ | Encryption Function |
| $SIG(M)$ | The message digital Signature |
| $PK_i$ | The public key of entity $i$ |
| $PR_i$ | The private key of an entity $i$ |
| $\delta_i$ | The entity's secret value |
| $\delta_G$ | The Group's secret value |
| $K_{VR}$ | The shared symmetric key between a vehicle and RUS |
| $K_G$ | The shared group key |
| $Msg_{ID}$ | The ID of the current group message |
| $SSV_i$ | The $i$th Secret Seed Value |

### Phase 1: Initiation

In this phase, we generate all the required group keys, along with their corresponding chains of N secret values. This ensures that all necessary secret values for group communication are pre-generated and readily available.

In the SMAD-LDS scheme, the communication time within a group will be divided into sessions. Each session has its group key and a chain of N secret values. Each secret value in the chain will be used once. When all secret values are used, the current session will be closed, and a new session will be started. For each session, RSU will choose random bits to be used as a symmetric key. This key will be used as the group key $K_G$. Moreover, it will choose a random secret seed value $SSV_i$ to create a chain of N secret values, a chain of N hashes, as shown in Fig. 2. These secret values in the chain, from back to front, will be used in signing the group's messages. These secret values will be used, in a novel and efficient way, to sign the group's messages. This is a completely novel idea inspired by blockchain.

### Phase 2: Registration

The SMAD-LDS scheme requires all vehicles to register with the RSU. This registration process prevents malicious vehicles from participating in group communication, including broadcasting and sending direct messages.

In the SMAD-LDS scheme, each RSU regularly broadcasts a beacon message every 100 ms. This message contains the $ID_R$, $TS$, the RSU's location coordinates $X_{pos}, Y_{pos}$, and The ID of the last message sent to the vehicles into its group $Msg_{ID}$.

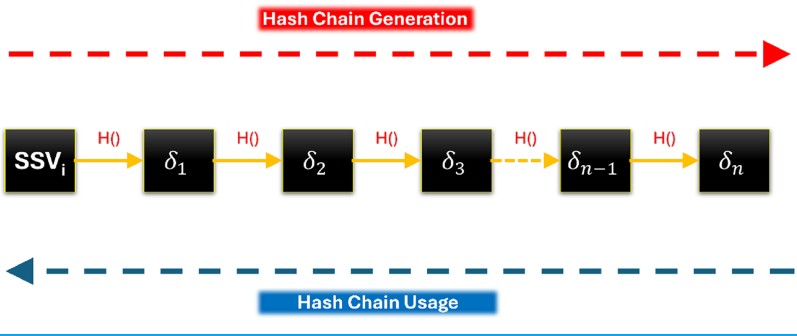

**Figure 2 Chain of Hashes.**

$$m = TS, Xpos, Ypos, ID_R, Msg_{ID}$$
$$Sig = E(H(m), PR_R) \tag{1}$$

When a vehicle enters a new region covered by an RSU, it will select a secret value $\delta_V$. Then, it encrypts both secret the value $\delta_V$ and its real identity $ID_v$ using the public key of the RSU $PK_R$. After that, it generates a message containing the current timestamp, the vehicle's location coordination $X_{pos}, Y_{pos}$, the vehicle's certificate $Cert_V$, and the previously encrypted data. Finally, it signs the message using its private key $PR_V$ and sends it to the RSU.

$$m = TS, Xpos, Ypos, Cert_V, ID_R, E((ID_v \parallel \delta_V), PK_R)$$
$$Sig = E(H(m), PR_V) \tag{2}$$

When the RSU receives the registration message, it will check the message's integrity and the vehicle's authenticity. Upon authenticating the vehicle, it will generate a new shared symmetric key as $K_{VR}$ besides a new vehicle's pseudo-identity $PID_v$ as in Eqs. (3) and (4), respectively:

$$K_{VR} = H(TS \parallel ID_v \parallel \delta_V). \tag{3}$$
$$PID_v = H(ID_v \parallel \delta_V). \tag{4}$$

The current group key $K_G$, the current group message ID ($Msg_{ID}$), and the current group's secret value $\delta_G$ are encrypted utilizing the previously generated shared symmetric key $K_{VR}$. After that, it will generate a response message that contains $ID_R, TS$, the RSU's location coordinates $X_{pos}, Y_{pos}$, and the previously encrypted data. Finally, it signs the message utilizing its private key $PR_R$. and sends it to the vehicle.

$$m = ID_R, TS, Xpos, Ypos, E((K_G \parallel Msg_{ID} \parallel \delta_G), K_{VR})$$
$$Sig = E(H(m), PR_R) \tag{5}$$

When the vehicle receives the registration's response message, it will check its integrity and authenticity. Then, it will calculate the shared symmetric key and its pseudo-identity as in Eqs. (3) and (4), respectively. After that, it will decrypt the encrypted data to get and store the current ID of the group message, the group's secret value, and the group's key. The message sequence is depicted in Fig. 3.

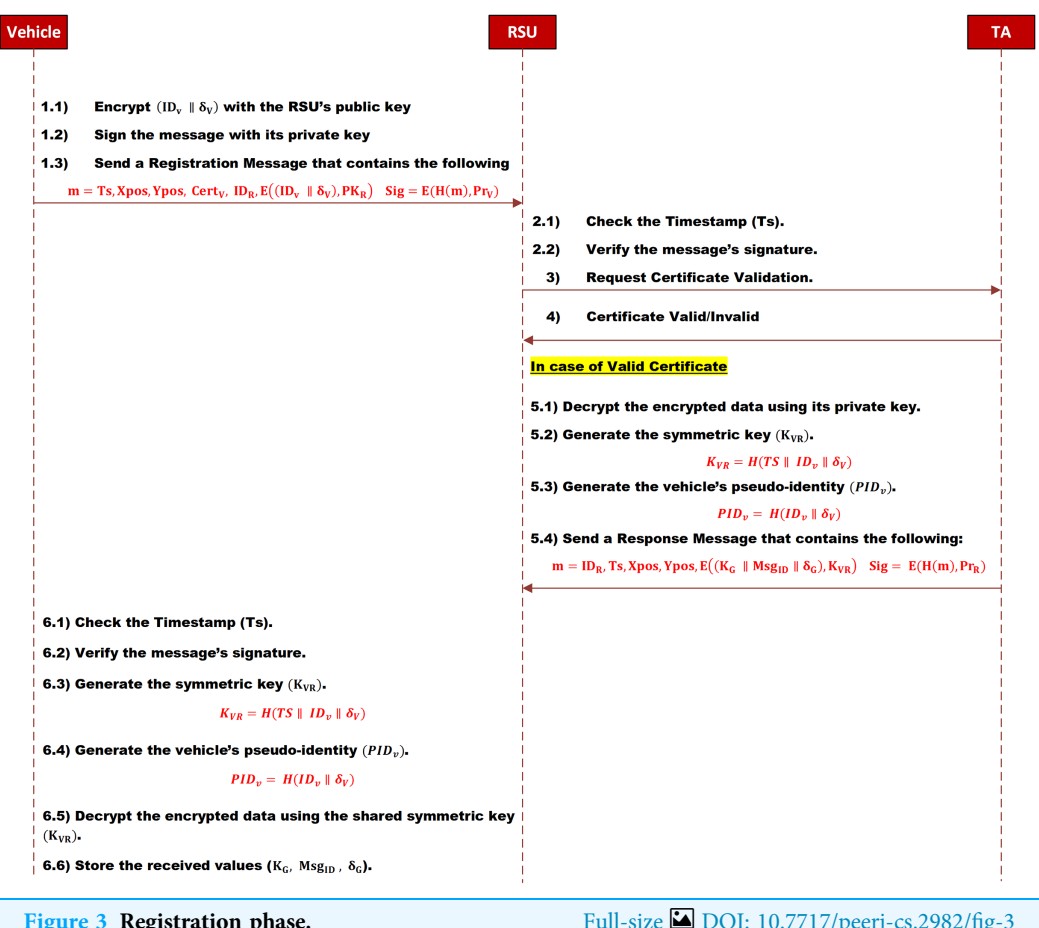

**Figure 3 Registration phase.**

### Phase 3: Send/receive messages

During this phase, the previously registered vehicles will be able to send and receive different types of messages. The SMAD-LDS scheme utilizes two distinct message types: group and one-to-one. As mentioned before, all messages in this scheme must be sent through the RSU.

### A. Send and receive a group message

When a vehicle wants to send data to a group, it will encrypt it using the shared secret key $K_{VR}$. Then, it will create a new message that contains the encrypted data, the current timestamp, the vehicle's location coordination $X_{pos}, Y_{pos}$, and the vehicle's pseudo-identity. Subsequently, a digital signature is appended to the message. Traditional message signing involves hashing the message content and encrypting the resulting hash with the originator's private key. This approach incurs substantial computation overhead for both signing and verification. SMAD-LDS addresses this limitation by employing a lightweight signing mechanism. In SMAD-LDS, the message is signed by hashing the concatenation of the message and the vehicle's secret value $(\delta_v)$. This signature technique provides a reduction in the computation overhead of signature generation and verification compared

to traditional methods. Note that the vehicle's secret value ($\delta_v$) is shared only between the vehicle and the RSU. Finally, it sends the message with its signature to the RSU. The message signature, in this work, is created by hashing the message with a secret value. Hence, there is no need to encrypt it.

$$m = TS, Xpos, Ypos, PID_V, \ ID_R, E(Data, K_{VR})$$
$$Sig = H(m \parallel \delta_v)$$
(6)

When the RSU receives the message, it will check the message's integrity and the vehicle's authenticity. This check is achieved by hashing the message with the vehicle's secret value ($\delta_v$) and verifying it against the message signature. If it is not valid, it will just drop it. Otherwise, if it is valid, it will decrypt the encrypted data using the shared symmetric key ($K_{VR}$). After that, it will encrypt it again by using the shared group key ($K_G$). Next, it will create a message that contains the encrypted data, the RSU's identity ($ID_R$), the current timestamp $TS$, and the next group's secret value ($\delta_{G-1}$). After that, it will sign the message by hashing the concatenation of the message and the current group's secret value ($\delta_G$). Finally, it broadcasts the message with its signature to the group.

$$m = ID_R, TS, Msg_{ID}, E(Data, K_G), \delta_{G-1}$$
$$Sig = \ H(m \ \parallel \delta_G)$$
(7)

Upon receiving the broadcast message, each vehicle within the group performs the following: First, it verifies the message's integrity and the RSU's authenticity. This check is achieved by hashing the message with the stored group's secret value ($\delta_G$) and verifying it against the message signature. If it is valid, it will hash the received ($\delta_{G-1}$) and then compare the result against the stored group's secret value ($\delta_G$). If it is not valid, it will drop the message. Otherwise, it will update the stored group's secret value ($\delta_G$) with the new received value ($\delta_{G-1}$). Finally, it will decrypt the encrypted data using the shared group key ($K_G$). The message sequence is depicted in Fig. 4.

### B. Send and receive a one-to-one message

When a vehicle ($V_1$) wants to send data to a vehicle ($V_2$), it will encrypt it using the shared secret key ($K_{V_1R}$). Then, it will create a new message that contains the encrypted data, the current timestamp, the vehicle's location coordination $X_{pos}, Y_{pos}$, the vehicle's pseudo-identity ($PID_{V_1}$), and the destination vehicle's pseudo-identity ($PID_{V_2}$). After that, it will sign the message by hashing the concatenation of the message and the vehicle's secret value ($\delta_{V_1}$). Finally, it sends the message with its signature to the RSU.

$$m = TS, Xpos, Ypos, PID_{V_1}, PID_{V_2}, ID_R, E(Data, K_{V_1R})$$
$$Sig = H(m \parallel \delta_{V_1})$$
(8)

When the RSU receives the message, it will check the message's integrity and the vehicle's authenticity. If it is valid, it will decrypt the encrypted data using the shared symmetric key ($K_{V_1R}$). After that, it will encrypt it again by using the shared key between the RSU and the destination vehicle ($K_{V_2R}$). Next, it will create a message that contains the

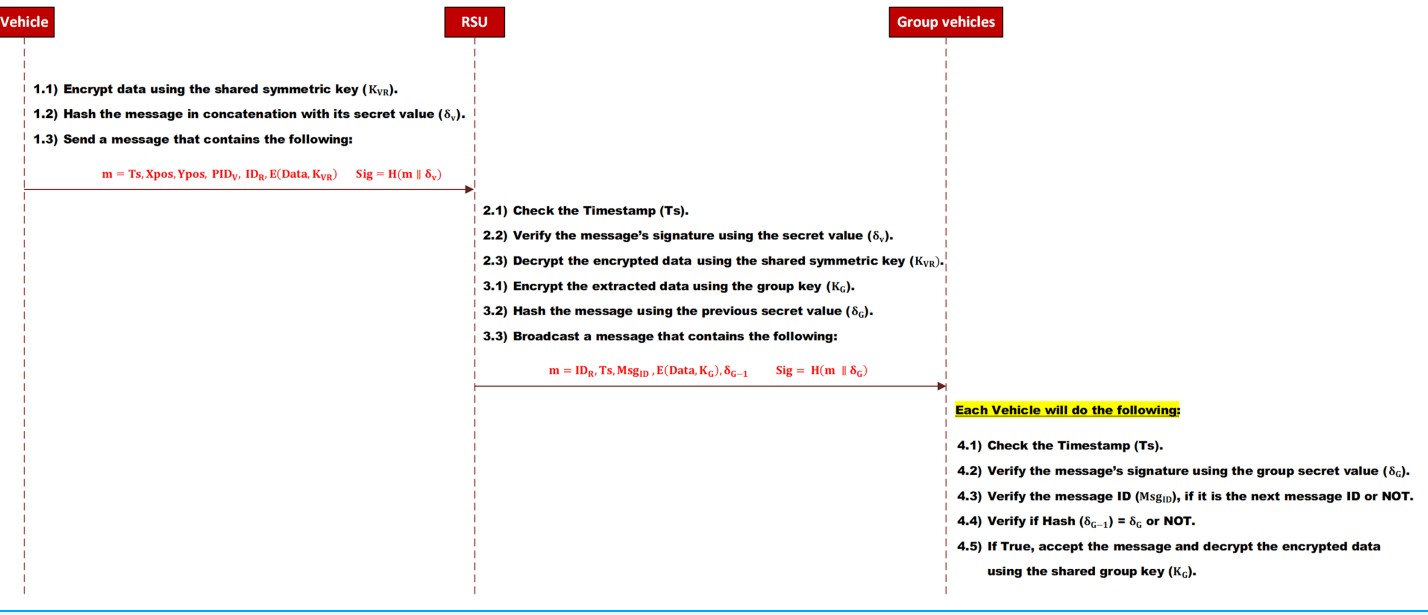

**Figure 4  Send a group message.**

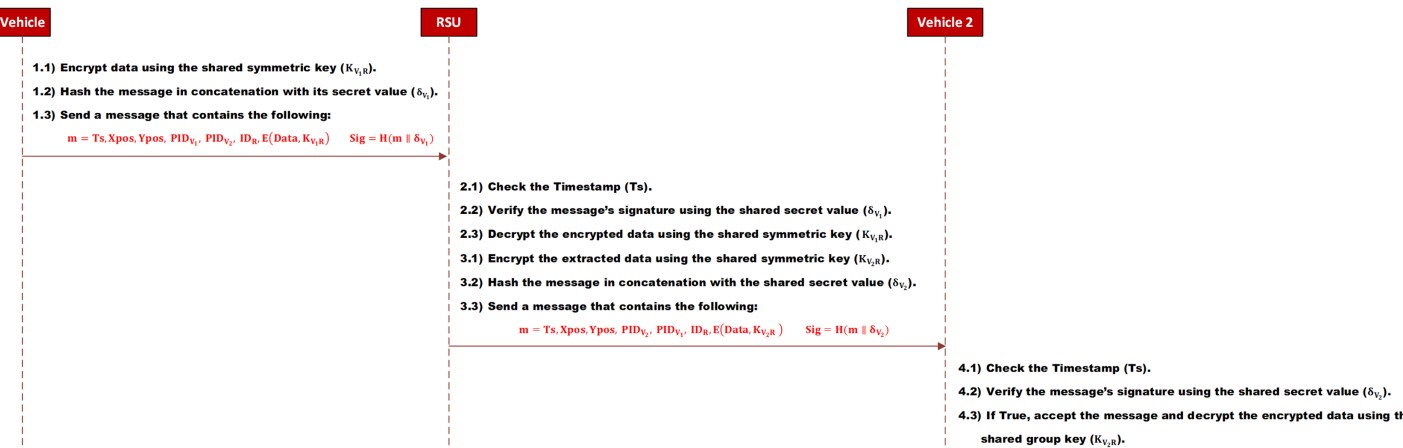

**Figure 5  Send a one-to-one message.**

encrypted data, the RSU's identity ($ID_R$), the current timestamp *TS*, and the pseudo-identity of both vehicles. After that, it will sign the message by hashing the concatenation of the message and the secret value of the destination vehicle ($\delta_{V_2}$). Finally, it sends it to the destination vehicle.

$$m = TS, Xpos, Ypos, PID_{V_2}, PID_{V_1}, ID_R, E(Data, K_{V_2R})$$
$$Sig = H(m \parallel \delta_{V_2})$$
(9)

When the destination vehicle receives the message, it will verify the message's integrity and the RSU's authenticity. If the message is valid, it will be accepted; otherwise, it will be dropped. The message sequence is depicted in Fig. 5.

*Phase 4: Group key update*

In the SMAD-LDS scheme, the group key is shared among all vehicles within the same group. Thus, if the key is exposed, an attacker could access the group's communications until the key is updated. To maintain robust security and ensure secure message dissemination within the group, the RSU regularly updates the group key. This process involves identifying the specific situations that trigger a group key update and then replacing the current key with a pre-generated one. In the SMAD-LDS scheme, the group key will be updated based on the following situations:

- **Exceeding the number of encrypted/decrypted messages (N messages).** In this case, the group key will be updated after all values in the current chain of secret values are used (end of session). As mentioned before, the RSU updates the active group key and the chain of secret values at the beginning of each session.
- **Exceeding a specified time interval**. In this case, the RSU will update the group key after a specified time interval, even if it is not used frequently. Moreover, the time interval is proportional to the key length.
- **Detecting a malicious vehicle.** If the RSU detects and revokes a malicious vehicle, it will update the group key to ensure secure message dissemination within the group.

  There are two different methods to update the group key as follows:

- **Key encapsulation method (KEM).** In this method, the RSU will share the new group key by sending it individually to each vehicle within the group. Consequently, the RSU will encrypt the new key using the public key of the vehicle. This method is suitable for key updates when exceeding a specified time interval or detecting a malicious vehicle.
- **Key derivation method (KDM).** In this method, the new key is derived from the old one. The derivation is done by hashing the old group key with some input parameters, such as a counter or a label, to generate the new group key. This method is suitable for key updates when the number of encrypted/decrypted messages (N messages) is exceeded.

## PERFORMANCE ANALYSIS

As previously mentioned, the growing number of message transmissions in the IoV, driven by its diverse communication types, necessitates an efficient and timely message dissemination scheme within a potentially congested network. Therefore, minimizing computation and communication overheads is critical. This reduction alleviates traffic congestion, consequently enhancing network latency and success rates. Additionally, the scheme utilizes a lightweight technique, which ensures minimal memory usage and does not impose any significant memory overhead. Consequently, evaluating the effectiveness of any message dissemination scheme requires an examination of the delivery time, which is directly impacted by both types of overhead. This section will analyze both the computation and communication overheads incurred by the proposed SMAD-LDS scheme during each of its phases. Furthermore, the performance of the SMAD-LDS

scheme will be compared against existing schemes in the literature. Both the proposed and existing schemes were implemented and tested on a Windows 10 machine with a 2.60 GHz Intel Core processor and 8 GB of RAM. The results demonstrate the efficiency of the SMAD-LDS scheme in terms of both computation and communication overhead.

## Computation overhead

In this section, we evaluate and compare the computation overhead of the SMAD-LDS scheme to other existing schemes. To assess the computational overhead introduced by the SMAD-LDS scheme, we employed a simulation methodology similar to those described in *Mistareehi & Manivannan (2022)* and *Naskar et al. (2024)*. This evaluation focuses on the cryptographic operations within our scheme, measuring their average execution times to determine the computational overhead. The average execution time of all cryptographic operations used in the SMAD-LDS scheme and the literature's schemes is listed in Table 2. We compared our computation overhead results with other existing schemes based on the total execution time of the required cryptographic operations. In our comparisons, we ensure that each cryptographic algorithm provides an equivalent level of security to ensure a fair comparison. Hence, we used RSA with a key length of 2048, which approximately has the same security level as ECC with a key length of 256 and AES with a key length of 256 (*Mahto & Yadav, 2017*).

### The computation overhead of the registration phase (Phase 2)

In the registration phase, the total computation time required to complete the group registration is examined and analyzed. In the SMAD-LDS scheme, it takes $3T_{E\_ECC} + 3T_{D\_ECC} + T_{E\_AES} + T_{D\_AES} + 4T_H \approx 4.03$ ms to complete a group registration. Hence, our scheme provides less computation time by 92.4% and 46.8% compared to schemes in *Mistareehi & Manivannan (2022)* and *Mohamed, Ahmed & Sadek (2021)*, respectively. The total time required to perform the cryptographic operations in the proposed scheme and literature schemes is listed in Table 3.

### The computation overhead of sending and receiving a message (Phase 3)

In this phase, we examine the total computation time required to send and receive a group message. In the SMAD-LDS scheme, it requires $2T_{E\_AES} + 2T_{D\_AES} + 5T_H \approx 0.445$ ms to send and receive a group message. The computation time in the SMAD-LDS scheme is enhanced by 98% and 94% compared to schemes in *Mistareehi & Manivannan (2022)* and *Mohamed, Ahmed & Sadek (2021)*, respectively. Moreover, we examine the computation time required to send and receive a one-to-one message. In the SMAD-LDS scheme, it required $2T_{E\_AES} + 2T_{D\_AES} + 4T_H \approx 0.422$ ms to send and receive a one-to-one message. The computation time in our scheme is less by 43.9% compared to the result in the scheme (*Mohamed, Ahmed & Sadek, 2021*). The total time required to perform the cryptographic operations in the proposed scheme and literature scheme is listed in Table 4.

## Communication overhead

As mentioned earlier, the effectiveness of a message dissemination scheme depends on the total amount of data transmitted. To evaluate the communication overhead of our

**Table 2 The execution time of different cryptographic operations.**

| Cryptographic operation | Notation | Avg. execution time (ms) |
|---|---|---|
| Encryption using RSA (2048 bits) | $T_{E\_RSA}$ | $\approx 0.25$ |
| Decryption using RSA (2048 bits) | $T_{D\_RSA}$ | $\approx 12.95$ |
| Decryption using CRT-RSA (2048 bits) | $T_{D\_CRTRSA}$ | $\approx 6.73$ |
| Encryption using ECC (256 bits) | $T_{E\_ECC}$ | $\approx 0.15$ |
| Decryption using ECC (256 bits) | $T_{D\_ECC}$ | $\approx 1.11$ |
| Encryption using AES (256 bits) | $T_{E\_AES}$ | $\approx 0.13$ |
| Decryption using AES (256 bits) | $T_{D\_AES}$ | $\approx 0.038$ |
| One-way Hash Function | $T_H$ | $\approx 0.023$ |

**Table 3 Comparison of computation cost for phase 2.**

| Scheme | Computation cost | Total time (ms) |
|---|---|---|
| *Mistareehi & Manivannan (2022)* | $4T_{E\_RSA} + 4T_{D\_RSA} + 4T_H$ | $\approx 52.90$ |
| *Mohamed, Ahmed & Sadek (2021)* | $T_{E\_RSA} + T_{D\_CRTRSA} + 3T_{E\_AES} + 3T_{D\_AES} + 4T_H$ | $\approx 7.57$ |
| SMAD-LDS | $3T_{E\_ECC} + 3T_{D\_ECC} + T_{E\_AES} + T_{D\_AES} + 4T_H$ | $\approx 4.03$ |

**Table 4 Comparison of computation cost for phase 3.**

| Message type | Scheme | Computation cost | Total time (ms) |
|---|---|---|---|
| Group message | *Mistareehi & Manivannan (2022)* | $2T_{E\_RSA} + 2T_{D\_RSA} + 2T_{E\_AES} + 2T_{D\_RSA} + 4T_H$ | $\approx 26.82$ |
| | *Mohamed, Ahmed & Sadek (2021)* | $T_{E\_RSA} + T_{D\_CRTRSA} + 3T_{E\_AES} + 3T_{D\_AES} + 4T_H$ | $\approx 7.57$ |
| | SMAD-LDS | $2T_{E\_AES} + 2T_{D\_AES} + 5T_H$ | $\approx 0.445$ |
| One-to-one message | *Mohamed, Ahmed & Sadek (2021)* | $4T_{E\_AES} + 4T_{D\_AES} + 4T_H$ | $\approx 0.752$ |
| | SMAD-LDS | $2T_{E\_AES} + 2T_{D\_AES} + 4T_H$ | $\approx 0.422$ |

proposed scheme, we calculate the total bytes sent and received between any two entities. This section evaluates and compares the communication overhead of the SMAD-LDS scheme with that of existing schemes. Initially, we assume the size of all fields inside the message as shown in Table 5.

### The communication overhead of the registration phase (Phase 2)

In this phase, to complete the vehicle registration process at the RSU, two messages are exchanged between them. To compute the communication overhead, we compute the total number of bytes sent and received between the vehicle and the RSU. In the SMAD-LDS scheme, it is required to exchange 1,096 bytes between a vehicle and the RSU. While it required 648 bytes and 1,528 bytes for the schemes in *Mohamed, Ahmed & Sadek (2021)* and *Mistareehi & Manivannan (2022)*, respectively. Accordingly, the communication overhead of our scheme is less efficient by 41% compared to the scheme (*Mohamed, Ahmed & Sadek, 2021*) and more efficient by 28% than the scheme (*Mistareehi & Manivannan, 2022*).

**Table 5 Size of required message fields.**

| Message field | Size (Bytes) |
|---|---|
| Timestamp | 4 |
| Xpos | 4 |
| Ypos | 4 |
| ID | 20 |
| PID | 32 |
| Hash Digits | 32 |
| Certificate | 200 |
| Sign message with AES | 32 |
| Sign message with RSA | 256 |
| Sign message with ECC | 32 |
| Message ID ($Msg_{ID}$) | 2 |
| Secret value ($\delta$) | 16 |

***The communication overhead of sending and receiving a message (Phase 3)***
In this phase, we compute the total number of bytes required to send and receive a group message. In the SMAD-LDS scheme, a vehicle sends a message with a size of 128 bytes to the RSU. Then, the RSU broadcasts a message with a size of 106 bytes into its group. Consequently, the overall communication overhead in this phase is 234 bytes. While it required 450 bytes and 788 bytes for the schemes in *Mohamed, Ahmed & Sadek (2021)* and *Mistareehi & Manivannan (2022)*, respectively. Accordingly, the communication overhead of our scheme is more efficient by 48% and 70% than schemes in *Mistareehi & Manivannan (2022)* and *Mohamed, Ahmed & Sadek (2021)*, respectively.

In addition, we examine the communication overhead of sending and receiving a one-to-one message. In the SMAD-LDS scheme, a vehicle sends a message with a size of 160 bytes to the RSU. Then, the RSU generates a message with a size of 160 bytes and sends it to the destination vehicle. Consequently, the communication overhead in this phase is 320 bytes. This communication overhead is approximately the same as in the scheme of *Mohamed, Ahmed & Sadek (2021)*, which requires 324 bytes.

## System efficiency and scalability analysis

In this subsection, we evaluate the system efficiency and scalability of the proposed scheme by analyzing key performance metrics critical to the IoV environment. Specifically, we examine how the SMAD-LDS scheme performs under increasing network loads by assessing scalability, energy consumption, latency, memory usage, and message dissemination success rate. These metrics collectively provide a comprehensive understanding of the scheme's ability to maintain reliable and timely communication while efficiently utilizing computational and communication resources. Through this analysis, we demonstrate that the SMAD-LDS scheme not only reduces overhead but also ensures robust performance in dense and dynamic vehicular networks, which is essential for the practical deployment of IoV applications.

### Scalability analysis

Scalability is a key challenge in IoV due to high vehicle mobility, dynamic topologies, and dense urban traffic. Increased vehicle density leads to congestion, communication delays, and higher computational demands, which degrade message dissemination performance. Our scheme addresses these issues by reducing communication overhead by 70% and computation overhead by 94%, effectively lowering network congestion and processing load. This dual reduction minimizes end-to-end latency and prevents bottlenecks, enabling the SMAD-LDS scheme to maintain low latency and high throughput, even under heavy traffic, ensuring robust and efficient message dissemination.

### Energy consumption issue

Energy consumption is a major concern in IoV environments, especially for battery-limited on-board units (OBUs). The SMAD-LDS scheme effectively reduces both communication and computation overheads, which are primary contributors to energy use. Lower communication overhead decreases the energy spent on wireless transmissions and receptions, while reduced computation overhead lessens the processing load and dynamic power consumption of OBUs by simplifying cryptographic and data operations. This combined reduction enhances energy efficiency, extending OBU battery life and supporting longer IoV deployments.

### Latency mitigation through overhead reduction

To address the primary concern of system latency, our methodology directly targets the fundamental sources of delay in IoV message dissemination systems by minimizing both the computational burden on individual nodes and the communication overhead between vehicles and infrastructure. The resulting approximate reductions of 94% in computation and 70% in communication overhead demonstrate our scheme's effectiveness in ensuring message dissemination occurs within acceptable timeframes for safety-critical vehicular applications.

### Memory utilization

Memory usage concerns in the IoV networks are addressed by both the inherent capabilities of modern vehicles, featuring sufficient power and sophisticated onboard computing, and our optimized scheme. The SMAD-LDS scheme includes a 94% reduction in computation overhead, which proportionally decreases buffer space needs. Furthermore, the implementation of lightweight ECC ensures minimal memory usage during cryptographic operations without compromising security.

### Improved success rate via reduced traffic and delay

The success rate of message dissemination, crucial for IoV safety, is enhanced by our overhead reduction scheme through decreased traffic and delay. Lower communication overhead reduces message size and network congestion, increasing the probability of delivery. Simultaneously, a 94% reduction in computation overhead speeds up processing and transmission, reducing timeouts and ensuring the timely delivery of critical messages.

By optimizing both processing and transmission, the SMAD-LDS scheme maximizes the likelihood of successful and prompt message reception in safety-critical scenarios.

# ANALYSIS OF SECURITY REQUIREMENTS AND ATTACK COUNTERMEASURES FOR THE SMAD-LDS SCHEME

The performance section demonstrates that the SMAD-LDS scheme provides improved and lightweight message dissemination, reducing computation and communication overheads. This lightweight design enables the SMAD-LDS to handle the growing number of vehicles and devices without system overload, thus effectively mitigating scalability issues while maintaining both security and performance. This section analyzes both the security requirements and attack countermeasures in detail.

## Comparative analysis of lightweight authentication schemes

To comprehensively evaluate the efficiency and security of the proposed SMAD-LDS scheme, we present a comparative analysis against several state-of-the-art lightweight authentication methods as shown in Table 6. This comparison focuses on critical performance metrics, including cryptographic techniques employed, execution time, signature size, and the level of security guarantees provided. By systematically comparing these parameters, we aim to highlight the distinctive advantages of SMAD-LDS in achieving a balanced trade-off between computational efficiency and robust security, which is crucial for practical deployment in the IoV environments.

## Security requirements

The proposed SMAD-LDS scheme satisfies all the following security requirements:

- **Message authentication with privacy preservation.** It is important to ensure the authenticity of vehicles besides preserving their privacy. To ensure that, each vehicle must register itself at the RSU. Subsequently, the RSU checks the legitimacy of the vehicle's certificate at the TA. If it has a valid certificate, it will accept it and allow it to join its group. Moreover, all communications in our system must be done through the RSU. So, when a vehicle wants to send anything to the group, it must send it to the RSU. Consequently, the RSU can verify the authenticity of the sender vehicle on behalf of all vehicles in its group.

- **Message integrity and non-repudiation.** The SMAD-LDS scheme ensures message integrity with non-repudiation in one operation. In this operation, we hash the message's bytes in concatenation with a secret value, as we mentioned in detail in our scheme. The hash function ensures the message's integrity. Moreover, the added secret value achieves the non-repudiation requirement. This secret value is only known by the sender vehicle.

- **Message confidentiality.** In our scheme, all messages are transmitted in an encrypted format. In the case of communication between a vehicle and an RSU, all messages will be encrypted using the shared key before their transmission. Therefore, there is no other vehicle that can get useful information from the encrypted message. In the case of

**Table 6 Comparison of SMAD-LDS with existing lightweight authentication schemes.**

| Criteria | | Mohamed, Ahmed & Sadek (2021) | Mistareehi & Manivannan (2022) | SMAD-LDS scheme |
|---|---|---|---|---|
| Entities involved | | Vehicles, RSUs, Trusted Authority (TA) | Vehicles, RSUs, Group Leaders (GLs), DMV (Trusted Authority) | Vehicles, RSUs, Trusted Authority (TA) |
| Types of messages | | Broadcast and One-to-One Messages | Broadcast Messages Only | Broadcast and One-to-One Messages |
| Signature scheme | | AES digital signatures over SHA-256 hashes | RSA digital signatures over SHA-256 hashes | Only SHA-256 hashes |
| Signature size | | 32 Bytes | 256 Bytes | 32 Bytes |
| Authentication | Registration phase | Symmetric key-based authentication with certificate checks | RSA signatures validated with DMV-issued certificates | Symmetric key-based authentication with certificate checks |
| | Send/receive messages phase | AES digital signatures over SHA-256 hashes | RSA digital signatures over SHA-256 hashes | Only SHA-256 hashes |
| Privacy approach | | Dynamic pseudonyms (PID) issued per region | Static pseudonyms (PID) with certified pseudo-IDs from DMV | Dynamic pseudonyms (PID) issued per region |
| Message verification | | Performed by RSU after receiving hashed and AES-encrypted messages | Performed hierarchically: GLs verify, aggregate and RSUs verify | Performed by RSU after receiving hashed and AES-encrypted messages |
| Overhead optimization | | • Use CRT-RSA as Public key cryptography.<br>• Use AES for message decryption and authentication | • Introduces Group Leaders (GLs) to share RSU overhead in dense regions | • Use Hashes for message signature and authentication.<br>• Use ECC as Public key cryptography.<br>• Use AES for message decryption and authentication |
| Security guarantees | | Authentication, confidentiality, integrity, privacy, non-repudiation, freshness | Authentication, confidentiality, integrity, privacy, non-repudiation, freshness | Authentication, confidentiality, integrity, privacy, non-repudiation, freshness, access control, key management |
| Replay attack protection | | ✓ | ✓ | ✓ |
| Impersonation attack protection | | ✓ | ✓ | ✓ |
| Sybil attack protection | | ✓ | ✓ | ✓ |
| Fake RSU attack | | ✓ | | ✓ |
| Traceability attack | | ✓ | | ✓ |
| Scalability | | Less scalable | Moderate scalable under high density | High scalable under high density |
| Latency | | High | High | Low |
| Memory usage | | High | High | Low |
| Successful rate | | Low | Low | High |
| Communication cost | | Medium | High | Low |

| Table 6 (continued) | | | | | |
|---|---|---|---|---|---|
| Criteria | | | Mohamed, Ahmed & Sadek (2021) | Mistareehi & Manivannan (2022) | SMAD-LDS scheme |
| Computation time (Encryption/ Sign) | Registration phase | Message encryption | Using CRT-RSA ≈ 0.25 ms | Using RSA ≈ 0.25 ms | Using ECC ≈ 0.15 ms |
| | | Message decryption | Using CRT-RSA ≈ 6.73 ms | Using RSA ≈ 12.95 ms | Using ECC ≈ 1.11 ms |
| | | Message signature | ≈ 0.273 ms | ≈ 0.273 ms | ≈ 0.173 ms |
| | | Signature verification | ≈ 6.753 ms | ≈ 12.973 ms | ≈ 0.061 ms |
| | Send/receive phase | Message encryption | Using AES ≈ 0.13 ms | Using AES ≈ 0.13 ms | Using AES ≈ 0.13 ms |
| | | Message decryption | Using AES ≈ 0.038 ms | Using AES ≈ 0.038 ms | Using AES ≈ 0.38 ms |
| | | Message signature | ≈ 0.153 ms | ≈ 0.153 ms | ≈ 0.023 ms |
| | | Signature verification | ≈ 0.061 ms | ≈ 0.061 ms | ≈ 0.046 ms |

sending a group message, the RSU will encrypt the message using the shared group key before sending it. Consequently, this message could be decrypted by the vehicles that joined the group only.

- **Message freshness.** In the SAND-LDS scheme, we include a timestamp in each disseminated message to ensure that the information they contain is recent and hasn't been replayed.
- **Access control.** The SMAD-LDS scheme divides the area into RSU-managed groups. Vehicles register with their local RSU to obtain access credentials, including the group key for secure communication. This registration is a key access control point for enforcing security policies, such as handling malicious vehicles. The RSU verifies the vehicle's certificate with a trusted authority during registration. If revoked, registration is denied by withholding the group key, preventing the vehicle from joining the group or participating in message dissemination. This ensures that revoked vehicles cannot decrypt messages or inject malicious data, preserving communication confidentiality, integrity, and authenticity within the group. Furthermore, periodic group key updates, securely distributed only to authorized vehicles, limit the access window for compromised vehicles.

## Security attacks and countermeasures

In the IoV, numerous security threats can affect its performance due to its communication nature (*Taslimasa et al., 2023*). These security risks must be encountered to ensure secure communications. The SMAD-LDS scheme could encounter and resist many types of well-known attacks. Here are some of them:

- **Modification attack.** This attack aims to modify the content of the transmitted message. The proposed work can prevent this attack by hashing the content of the message with a pre-defined secret value. Consequently, if an adversary intends to modify the message's content, it must first figure out the sender vehicle's secret value, which is very difficult.
- **Impersonation attack.** In this attack, a malicious vehicle pretends to be another legitimate vehicle. This attack is well-known as a Sybil attack or node impersonation

attack. The proposed scheme can encounter this attack by using a random pseudo-identity. This pseudo-identity is changed every time a vehicle joins a new group. Moreover, the malicious vehicle must know many secret parameters to pretend to be another vehicle. These secret parameters are such as the current secret value of the target vehicle, the shared key between it and the RSU, and the current pseudo-identity of the target vehicle.

- **Eavesdropping attack**. It represents a threat to message confidentiality, wherein an adversary seeks to gain unauthorized access to transmitted data. The proposed system mitigates this threat through the encryption of message content using a cryptographically secure secret key.

- **Fake RSU attack.** In this type of attack, an adversary pretends to be a legitimate RSU. The proposed scheme handles this type of attack through many steps. First, every vehicle is preloaded with the location, ID, and public key of all RSUs. Second, when a vehicle receives a one-to-one message from an RSU, it can authenticate it by hashing the message with the shared secret value and then comparing it with the included hash value. Third, in case of receiving a group's message, it will check it as we mentioned before by using the group secret value. In addition, the vehicle will verify the included $Msg_{ID}$ and the new group's secret value $\delta_{G-1}$ as follows:

  - The new $Msg_{ID}$ must equal the stored $Msg_{ID} + 1$.
  - The hashing of $\delta_{G-1}$ must equal the stored $\delta_G$.

- **Compromised RSU.** The SMAD-LDS scheme mitigates rogue RSU attacks through the preloading of RSU location, IDs, and public keys on vehicles, thereby preventing impersonation during authentication. Furthermore, periodic updates of RSU security parameters enhance system robustness. Nevertheless, the scenario of a compromised RSU, with its potential to cause data tampering, message injection, or denial of service, poses a considerable threat to IoV security and reliability. Addressing this specific vulnerability requires dedicated countermeasures, the development of which falls outside the scope of this article and is identified as critical future work.

- **Traceability attack.** This attack threatens vehicle privacy by aiming to reveal sensitive information about their movements and behaviors. Our approach counters this by employing randomly generated pseudo-identities for each vehicle, which are updated whenever a vehicle joins or leaves a group. This significantly complicates an attacker's ability to consistently track any vehicle. Nevertheless, authorized entities, such as RSUs and the TA, maintain the capability to disclose a vehicle's true identity for lawful investigations.

- **Replay attack.** In this attack, an attacker aims to resend a previously sent message without any changes to steal some information or for a reason. Our scheme is resistant to this type of attack, as each message sent in the network has a fresh timestamp. An attacker cannot modify or change the current message's timestamp without creating a new hash value with the new timestamp. Consequently, the attacker

must know the random secret value that is included in the original hash value, which is very difficult.

## CONCLUSIONS AND FUTURE WORKS

The Internet of Vehicles (IoV) relies on various applications designed to enhance driver safety. Therefore, secure and rapid message authentication and dissemination are crucial. This article introduces an enhanced secure message authentication and dissemination with lightweight digital signature (SMAD-LDS) scheme for the IoV, designed for efficient message exchange between vehicles and RSUs by minimizing communication and computation overheads. A novel, lightweight digital signature technique is employed for increased efficiency. Results demonstrate a significant reduction in computation costs for SMAD-LDS compared to existing solutions: at least 46.8% in the registration phase and at least 94% in the message send/receive phase. Communication costs are also improved, with reductions of 28% and 70% in the registration and message send/receive phases, respectively. Additionally, SMAD-LDS satisfies key security requirements, including entity and data authenticity, privacy preservation, and data confidentiality. The scheme's robustness against various attacks, such as modification, impersonation, eavesdropping, fake RSU, traceability, and replay attacks, is also validated.

Future works will concentrate on four key areas to enhance the SMAD-LDS scheme. The first area involves extending the scheme to support V2V direct communication, enabling more efficient information exchange among vehicles. The second area will address security concerns related to the single-point-of-failure issue and RSU compromise by enhancing the scheme's robustness in such scenarios. The third area will explore the potential of integrating recent quantum cryptography algorithms into the IoV architecture and analyze their impact on security and performance. Finally, we will examine and analyze various existing techniques for detecting malicious vehicles and revoking their certificates.

### Funding
The authors received no funding for this work.

### Competing Interests
Rowayda A. Sadek is an Academic Editor for PeerJ.

### Author Contributions
- Islam Z. Ahmed conceived and designed the experiments, performed the experiments, analyzed the data, performed the computation work, prepared figures and/or tables, authored or reviewed drafts of the article, and approved the final draft.
- Yasser Hifny analyzed the data, authored or reviewed drafts of the article, and approved the final draft.

- Rowayda A. Sadek conceived and designed the experiments, analyzed the data, authored or reviewed drafts of the article, and approved the final draft.

## Data Availability

Code is available in the Supplemental Files.

## Supplemental Information

Supplemental information for this article can be found online at http://dx.doi.org/10.7717/peerj-cs.2982#supplemental-information.

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
