# Peer review of "SMAD-LDS: enhanced secure message authentication and dissemination with lightweight digital signature in the Internet of Vehicles"

_PeerJ Computer Science, doi:10.7717/peerj-cs.2982_

## Round 0.1 · original submission · Major Revisions

· Academic Editor

Major Revisions

Please respond in detail to all the comments from the reviewers.

Reviewer 1 ·

Basic reporting

The authors have proposed an SMAD-LDS model for Internet of Vehicles (IoV) networks. However, the current version of the manuscript contains several shortcomings that hinder its suitability for publication. The primary concerns include the limited and unclear contributions of the work. The manuscript does not convincingly demonstrate any substantial advancements or original contributions to the field. Additionally, the results that support the findings are insufficient and lack concrete evidence. The paper includes only numerical data without adequate explanation or justification of how these values validate the proposed model.

Moreover, the manuscript’s structure and layout do not align with the standard format expected for research articles, making it challenging to follow and assess its scientific rigour. The literature review is also minimal, and several technical terms are inconsistently used, suggesting a lack of clarity in terminology.

In summary, the manuscript, in its current form, does not meet the standards for publication. It is strongly recommended that the authors conduct a comprehensive review of the manuscript, address these issues, and provide a more thorough research presentation, including more apparent contributions, well-supported results, a structured format, and consistent terminology.

Experimental design

There are no experimental design.

Validity of the findings

The findings are limited and not evident.

Reviewer 2 ·

Basic reporting

This paper introduces a robust message authentication scheme called SMAD-LDS, which features a lightweight digital signature technique. The scheme comprises four phases: initiation, registration, message transmission, and key updating.

Some questions?

1. What specific measures are taken in the SMAD-LDS scheme to ensure secure message authentication and dissemination?

2. Can you explain the significance of each of the four phases in the SMAD-LDS scheme: initiation, registration, send/receive message, and key updating?

3. How does SMAD-LDS's lightweight digital signature technique differ from traditional authentication methods?

4. What types of attacks is SMAD-LDS designed to resist, and how does it mitigate each threat?

5. Can you provide examples of scenarios where the IoV's enhanced driver experience significantly impacts safety?

6. What are the potential implications if vehicles within the IoV are not properly authenticated?

The Steps algorithm should be presented in a good way.

Add refs:
https://www.sciencedirect.com/science/article/pii/S0167404824000932
https://www.mdpi.com/1999-4893/16/8/381

Experimental design

-

Validity of the findings

-

Additional comments

-

·

Basic reporting

No comment.

Experimental design

The data collection process should be rigorously outlined, with attention to potential biases and confounding variables. The authors should demonstrate that they have taken steps to minimize these issues.

Validity of the findings

How well do the results contribute to advancing knowledge in the field?

Additional comments

Overall, this manuscript demonstrates a high level of scholarship and contributes significantly to the field, making it a valuable addition to the literature.

Reviewer 4 ·

Basic reporting

no comment

Experimental design

no comment

Validity of the findings

no comment

Additional comments

This work addresses the challenges of secure and efficient communication in Internet of Vehicles (IoV) networks by proposing SMAD-LDS, a novel lightweight digital signature scheme. SMAD-LDS is structured around four phases: initiation, registration, message exchange, and key update. The authors' evaluation shows substantial reductions in computational and communication costs across these phases, compared to several related methods. Furthermore, the scheme is shown to be resilient to a variety of security threats. The authors provide a useful way for the development of IoV networks. I will recommand the publication of this manuscript if the following comments are considered.

(1) The paper uses the term "lightweight digital signature." While intuitively understandable, a precise definition of what constitutes "lightweight" in this context (e.g., computational complexity, communication overhead, key size) should be provided early in the paper, preferably in the introduction.

(2) The introduction could benefit from a more comprehensive overview of existing authentication schemes in IoV. Currently, the introduction briefly touches upon the general security needs and challenges, but lacks a structured discussion of existing approaches and their limitations.

(3) some introduction of quantum digital signature (QDS) will inspire the further thinking about the possible research topics of utilizing quantum information technology in real life. The recent progress of QDS like [Phys. Rev. Appl. 20, 044011 (2023)] and the application of QDS in further area like [Sci. Adv. 10, eadk3258 (2024); Sci. Adv. 10, eadp2877 (2024)] may be helpful.

(4) The performance analysis focuses on computational and communication overhead comparisons. However, a detailed security analysis of SMAD-LDS, addressing potential vulnerabilities beyond the listed attacks, would strengthen the paper. For example, the paper should explicitly address issues such as key management, resilience against compromised RSU, and scalability for a large number of vehicles and RSUs. A discussion of the trade-offs between security and efficiency would also be beneficial.

Reviewer 5 ·

Basic reporting

no comment

Experimental design

no comment

Validity of the findings

no comment

Additional comments

In this paper, a secure message authentication and dissemination with a novel lightweight digital signature technique (SMAD-LDS) is proposed. The results results demonstrate that the computation cost in SMAD-LDS is reduced by at least 46.8% in comparison with other works in the literature. Also, the computation overhead, in the send/receive messages phase in SMAD-LDS, is reduced by at least 94%. In addition, the overall cost of communications in the proposed paper seems to have improved. The scheme seems to be resistant to many known attacks as well. I would like to recommend this paper with few minor concerns as given below.

1. There are many short sentences in the paper that must be combined into one sentence for better perceiving the message. For instance, in the abstract the two sentence s are: Moreover, many security requirements should be satisfied. Security requirements such as entity and data authentication, privacy preservation, and confidentiality. I would suggest revise the paper with a particular focus on grammatical and conjunction errors.
2. There is threat/attack model in the scheme. The authors are required to add this model in the preliminaries or analysis section.
3. I would suggest to reorganize the scheme by placing analysis section before performance evaluation section, while the security requirements may also be taken into preliminary section beside other details.
4. The figures 5, 6 and 7 appear to be a little vague, and can be further improved.
5. Please identify your future work, if any, in the conclusion section.

---

## Round 0.2 · Major Revisions

· Academic Editor

Major Revisions

Although 3/4 reviewers recommended acceptance (but without any specific comment), we stick to the general evaluation and remarks of Reviewer 1. Thus, we ask the authors try to improve their work and manuscript and resubmit it addressing *all of the 7 points with sub-items* enumerated by Reviewer 1.

Reviewer 1 ·

Basic reporting

The paper presents a well-structured authentication scheme (SMAD-LDS) for IoV, but it lacks key discussions on scalability, security modelling, key management, and quantum security. Addressing these missing points will significantly improve the manuscript’s impact and credibility. The comments are given below.

1. The paper claims that SMAD-LDS introduces a novel lightweight digital signature technique but does not provide a detailed comparison with existing lightweight cryptographic methods.
Missing Content: A structured comparison table with metrics (e.g., execution time, signature size, security guarantees) between SMAD-LDS and other lightweight authentication schemes.

2. The performance evaluation focuses mainly on computation and communication overheads but does not address scalability in high-traffic IoV environments. The effect of many vehicles and RSUs on system performance (latency, congestion, and key management).

3. The paper assumes that RSUs are fully trusted without discussing what happens if an RSU is compromised. The missing contents are the possible threat scenarios (e.g., rogue RSU, internal attacks) and the Mitigation strategies (e.g., multi-factor authentication for RSU, distributed trust models, blockchain-based RSU verification).

4. The paper introduces a group-based key management scheme but does not explain how to handle revoked or compromised vehicles.

5. The paper focuses on computational efficiency but does not address energy consumption, critical in IoV environments with battery-powered OBUs.

6. The paper evaluates computation time and communication overhead but does not justify why these specific metrics were chosen. Explanation of why other metrics (e.g., latency, memory usage, success rate) were not considered.

7. The authors miss some important and recent publications relevant to the work. It is suggested that these works be mentioned in the related work section and their major differences be shown. The suggested references are:

a. A Comprehensive Survey on Mobility-Aware D2D Communications: Principles, Practice and Challenges," in IEEE Communications Surveys & Tutorials, vol. 22, no. 3, pp. 1863-1886, thirdquarter 2020, doi: 10.1109/COMST.2019.2923708.

b. EEMDS: An Effective Emergency Message Dissemination Scheme for Urban VANETs. Sensors 2021, 21, 1588. https://doi.org/10.3390/s2105158

c. Secure Internet of Vehicles (IoV) With Decentralized Consensus Blockchain Mechanism," in IEEE Transactions on Vehicular Technology, vol. 72, no. 9, pp. 11227-11236, Sept. 2023, doi: 10.1109/TVT.2023.3268135.

d. AAKE-BIVT: Anonymous Authenticated Key Exchange Scheme for Blockchain-Enabled Internet of Vehicles in Smart Transportation," in IEEE Transactions on Intelligent Transportation Systems, vol. 24, no. 2, pp. 1739-1755, Feb. 2023, doi: 10.1109/TITS.2022.3220624

Experimental design

See the above comments.

Validity of the findings

See the above comments.

Additional comments

See the above comments.

Reviewer 2 ·

Basic reporting

Thank you for your revised version

Experimental design

Thank you for your revised version

Validity of the findings

Thank you for your revised version

·

Basic reporting

no comment

Experimental design

no comment

Validity of the findings

no comment

Reviewer 5 ·

Basic reporting

no comment

Experimental design

no comment

Validity of the findings

no comment

Additional comments

The authors have satisfactorily resolved all of my concerns. Therefore, I suggest this article for publication in its current form.

---

## Round 0.3 · Major Revisions

· Academic Editor

Major Revisions

I am sorry, but the version 2 of your manuscript still lacks the improvements required according to the remarks and suggestions of Reviewer 1.

---

## Round 0.4 · accepted · Accept

· Academic Editor

Accept

Reading through your manuscript by myself, I confirm that all the major concerns/criticisms of the reviewers have been well addressed in the 3rd version of your manuscript.

Congratulations, and thank you for your patience.